

# The effect of journal guidelines on the reporting of antibody validation

Joyce M. Hoek[1,2], Wytske M. Hepkema[2] and Willem Halffman[2]

[1] Department of Psychology, University of Groningen, Groningen, The Netherlands
[2] Institute for Science in Society, Radboud University Nijmegen, Nijmegen, The Netherlands

## ABSTRACT

**Background**. Despite the widespread use of antibodies as a research tool, problems with specificity, lot-to-lot consistency and sensitivity commonly occur and may be important contributing factors to the 'replication crisis' in biomedical research. This makes the validation of antibodies and accurate reporting of this validation in the scientific literature extremely important. Therefore, some journals now require authors to comply with antibody reporting guidelines.

**Methods**. We used a quasi-experimental approach to assess the effectiveness of such journal guidelines in improving antibody reporting in the scientific literature. In a sample of 120 publications, we compared the reporting of antibody validation and identification information in two journals with guidelines (*Nature* and the *Journal of Comparative Neurology*) with two journals without guidelines (*Science* and *Neuroscience*), before and after the introduction of these guidelines.

**Results**. Our results suggest that the implementation of antibody reporting guidelines might have some influence on the reporting of antibody validation information. The percentage of validated antibodies per article slightly increased from 39% to 57% in journals with guidelines, whereas this percentage decreased from 23% to 14% in journals without guidelines. Furthermore, the reporting of validation information of all primary antibodies increased by 23 percentage points in the journals with guidelines (OR = 2.80, 95% CI = 0.96-INF; adjusted $p$ = 1, one-tailed), compared to a decrease of 13 percentage points in journals without guidelines. Fortunately, the guidelines seem to be more effective in improving the reporting of antibody identification information. The reporting of identification information of all primary antibodies used in a study increased by 58 percentage points (OR = 17.8, 95% CI = 4.8-INF; adjusted $p$ = 0.0003, one-tailed) in journals with guidelines. This percentage also slightly increased in journals without guidelines (by 18 percentage points), suggesting an overall increased awareness of the importance of antibody identifiability. Moreover, this suggests that reporting guidelines mostly have an influence on the reporting of information that is relatively easy to provide. A small increase in the reporting of validation by referencing the scientific literature or the manufacturer's data also indicates this.

**Conclusion**. Combined with the results of previous studies on journal guidelines, our study suggests that the effect of journal antibody guidelines on validation practices by themselves may be limited, since they mostly seem to improve antibody identification instead of actual experimental validation. These guidelines, therefore, may require additional measures to ensure effective implementation. However, due to the explorative nature of our study and our small sample size, we must remain cautious towards other factors that might have played a role in the observed change in antibody reporting behaviour.

Corresponding author
Joyce M. Hoek, j.m.hoek@rug.nl

## INTRODUCTION

Antibodies are veritable workhorses in biomedical research. Used to label specific molecules or antigens (generally proteins), they allow researchers to map biomolecular processes in the cell. Through techniques such as western blotting (WB), immunohistochemistry, or ELISA, the use of antibodies has become widespread in biological research.

However, binding antibodies to antigens is intricate and full of complications. Antibodies may lack specificity and their affinity for specific antigens may vary. Furthermore, their affinity may vary with experimental conditions (*Baker, 2015b*), for example when pH levels or reagents denature proteins, altering protein folding and thereby the epitopes to which antibodies bind. Among tens of thousands of antibodies on offer (largely from commercial suppliers), researchers need to identify precisely which antibodies are most suitable for their antigens of interest, under their precise experimental circumstances. To aid this selection, extensive support tools have been developed, such as the Antibodypedia, or the Antibodyregistry, which support the identification of antibodies with research resource IDs (RRIDs) (*Bandrowski et al., 2016*).

Nevertheless, even with the help of such resources, the practice of antibody use remains complex. While researchers may identify the correct antibody for their specific research purposes *in principle*, verifying that the correct antibodies are used *in practice* is a different matter. Antibodies may vary from batch to batch, suppliers may not always be able to guarantee relevant quality standards, or earlier mistakes may be obfuscated by relying on locally established routines, such as habits, experimental skills, and techniques passed on in a laboratory. If identification and validation information about antibodies is not reported accurately, the possibility of experimental replication is jeopardised, and subsequent research may be built on errors. In turn, this may lead to wasted research, missed opportunities for medical innovation, or even patient safety risks. Taken together, the costs involved may be considerable. In fact, some commentators suggest problems with antibody validation, or lack of validation information, may be an important contributing factor to the 'replication crisis' in biomedical research (*Freedman, Cockburn & Simcoe, 2015*).

For over a decade, various researchers have expressed concerns about insufficient antibody validation in biomedical research (*Baker, 2015a*; *Bordeaux et al., 2010*; *Saper & Sawchenko, 2003*). The challenges are considerable. One study validating over 5,000 commercial antibodies for the Human Protein Atlas (HPA) showed that half of these antibodies were not suitable for the specific immunohistochemistry application in the HPA. The researchers concluded that every application of antibodies requires application-specific validation (*Berglund et al., 2008*).

Advocates of tighter validation have suggested techniques and principles to ensure correct antibodies are used and to improve reporting of antibody validation information in publications, although there is no universal standard as yet (*GBSI, 2016*). At the very

least, these advocates suggest that researchers should report information that establishes the identity of antibodies used, via reference to RRIDs or supplier identifiers, including catalogue and batch number (*Vasilevsky et al., 2013*). While many researchers rely on the literature to establish application-specific validity of antibodies, actual validation testing avoids repetition of older mistakes. This testing should verify not just the identity, avoiding supply-line errors (such as misidentification during personal exchange, or transport via suppliers), but also the antibody's specificity for the target protein and sensitivity in the specific application. Relatively simple checks include staining a western blot to check whether antibodies recognise antigens of the correct molecular weight, omitting the primary antiserum, and performing pre-adsorption controls (*Saper & Sawchenko, 2003*). While these methods might be useful as a first indicator of an antibody's specificity, they are not very stringent, and antibodies might still be found to be nonspecific upon more thorough validation (*Andersson et al., 2017*; *Bordeaux et al., 2010*; *Hewitt et al., 2014*; *O'Hurley et al., 2014*).

The International Working Group of Antibody Validation (IWGAV) proposed a very stringent validation procedure in 2016, with five 'pillars' for application-specific validation. These include genetic strategies (testing the antibody in conditions when the protein is not expressed), orthogonal strategies (comparing results for varying amounts of target protein identified by other means), independent antibody strategies (comparing results with alternative antibodies), expression of tagged proteins (using affinity tags or fluorescent proteins), and immunocapture followed by mass spectrometry. The IWGAV advises researchers to carry out at least one of these five methods in order to validate whether an antibody is truly specific to the application at hand. Furthermore, the IWGAV advises suppliers to also use at least one of these five pillars for validation, including validation for each new batch of antibodies, and to provide specific information on optimal use (*Uhlén et al., 2016*).

Several initiatives encourage researchers to improve antibody validation. Apart from calls to action in editorials, the development of databases, and improved validation by suppliers, some journals have also stepped up to the plate. The first and most vocal initiative was taken by the *Journal of Comparative Neurology* (*JCN*), introducing explicit requirements for antibody validation in its author instructions in 2003 (*Saper, 2005*; *Saper & Sawchenko, 2003*). Several journals followed suit and now require various levels of antibody validation information: some require only identification information; others require documentation or actual experimental proof of application-specific validity (*Steve et al., 2018*; *Gore, 2013*; *Nature, 2013*).

In essence, these journal initiatives constitute rule-based or 'regulatory' policy, attempting to change researchers' behaviour through regulation. These vary from insistent but voluntary guidelines, through checklists that attempt to 'nudge' authors into compliance (*Babic et al., 2019*), or actual rules at penalty of editorial rejection. In policy sciences, the track record of rule-based behaviour modification is mixed, at best. Rules require policing, which is expensive, and rules tend to fail if they lack community support, for example if they are perceived as meaningless or ineffectual. In matters of

research integrity, rules that blatantly diverge from actual practice have been shown to induce cynicism among researchers, decreasing willingness to comply (*Clair, 2015*).

Although journals have long been considered powerful 'gatekeepers of science' (*Crane, 1967*), and journals are generally expected to maintain standards in research communities, their power to do so is not absolute. Journals are also dependent on research communities for submitted articles and willingness of reviewer cooperation, compounded by the hard economics of publishing in which volume is often the backbone of the business model (*Larivière, Haustein & Mongeon, 2015*). Hence, the possibility of journals to raise standards may be limited by what the research community is willing to maintain.

From this perspective, it is interesting to investigate the effect of journal antibody guidelines on the presence and quality of reported validation information. For these purposes, we chose to compare antibody validation information in journals with guidelines and in equivalent journals without guidelines, i.e., a control group. Our hypothesis was that antibody validation information would improve in the journals that introduced antibody validation guidelines.

Before we turn to the details of our research method, we want to discuss some key methodological assumptions in validation testing, which touch upon quite fundamental issues of philosophy of science. First, we want to make an important distinction between antibody reliability and antibody validity. Antibodies may reliably produce the same results if experimental conditions are reproduced exactly, but if these conditions are based on misconceptions, they could simply reliably reproduce systematic error. A striking example of how the reproduction of error might lead research astray, comes from the recent evaluation of antibodies against oestrogen receptor beta in breast cancer research. *Andersson et al. (2017)* discovered that the antibodies most cited in the literature had systematically stained the wrong proteins in tissue samples and that the target protein was not even present in breast cancer tissue, affecting two decades of research. Hence, validation would ideally constitute the experimental confirmation under different conditions, providing more robust replication, rather than identical reproduction, as recent philosophy of science has argued (*Leonelli, 2018*). In the case of antibodies, the principles of the 'five pillars' offer such variation in experimental conditions. In contrast, strictly speaking, reference to exact reproduction of antibodies as used in the literature only constitutes proof of reliability, although it is frequently presented as proof of validity.

Second, while exact reproduction can be considered important evidence for antibody reliability in specific applications, in the practice of antibody research this evidence is not absolute: it always remains possible to question whether the reproduced experiment was indeed identical or performed equally expertly. Lack of exact reproduction raises the possibility that the repeat experiment failed to reproduce the exact original conditions and materials, or lacked some quality in the performer, such as particular experimental skills (*Collins, 1985*). In the case of antibodies, this is particularly pertinent in the case of polyclonal antibody batches that have run out, for example as source animal populations are terminated. Moreover, whether an application of antibodies constitutes an exact reproduction may be difficult to establish based on concise information provided in databases or even publications. Lack of reproduction therefore does not automatically
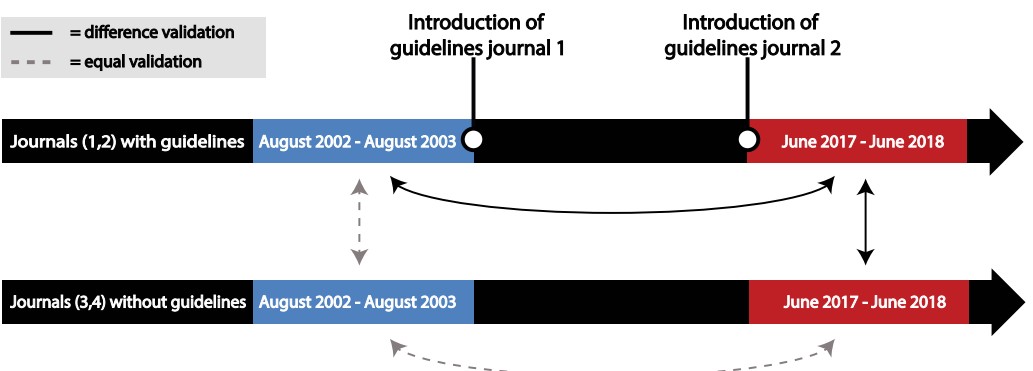

**Figure 1** **Overview of the research strategy.** Samples of 15 articles were taken from journals that did (1 and 2) and that did not (3 and 4) introduce antibody reporting guidelines. These samples were taken from a timeframe before (August 2002–August 2003) and after (June 2017–June 2018) the introduction of guidelines by journal 1 and 2. An increase in antibody validation reporting was expected between the samples connected with a black arrow, while no difference was expected between the samples connected with a dashed grey arrow.

disqualify the reliability of the original experiment and exact reproduction may be hard to assess in practice.

In scoring the quality of validation information, we should therefore pay attention to information that establishes the identity of antibodies, to evidence for antibodies' reliability in the specific intended purposes, and to evidence that experimentally confirms the antibodies' adequacy to identify the intended antigens with precision. While only the latter constitutes proper validation, information about identity and reliability is a precondition of such higher levels of validation.

## MATERIALS & METHODS

### Research strategy

In this exploratory study, we used a quasi-experimental study design to assess whether journal antibody validation guidelines can improve antibody validation reporting (see Fig. 1). We selected two journals that had adopted such guidelines and two journals with a comparable impact factor and research area, but without guidelines. We took random samples of 15 articles from each journal from a one-year timeframe before and after the two journals implemented antibody reporting guidelines. We used the study performed by *Han et al. (2017)* as an indication of the sample size and did not perform a power calculation beforehand. Since the first selected journal with guidelines created awareness of the antibody problem in August 2003, the 'before guidelines' samples consisted of articles that were published between August 2002 and August 2003. The 'after guidelines' samples consisted of articles that were published between June 2017 and June 2018, because the second journal with guidelines implemented its most stringent guidelines in June 2017. We then used an antibody validation coding table to assess the antibody validation information provided by each of the articles. We evaluated a total of 120 articles.

[1]Editorial policies at *JCN* have recently changed. The Antibody Characterization section in force at the time of this study is no longer a requirement.

## Journal selection

### *Intervention group*

The *Journal of Comparative Neurology (JCN)* and *Nature* were selected as the two journals with antibody validation guidelines. With regard to antibodies, *JCN* requires authors to add an 'Antibody Characterization' section to the methods section of a paper.[1] This section should include complete identification information of the used antibodies and proof of their validity. Authors can report their own validation or describe and cite evidence from the literature. The guidelines further specify that just a casual reference to a supplier's data sheet does not count as reliable evidence of specificity (information from Author Guidelines section of website *Journal Comparative Neurology,* retrieved on 04-05-2018). When authors want to submit a manuscript to *JCN,* they need to check a box that states that their publication provides full (validation) information on all antibodies used (article submission process tested on 09-05-2018).

The journal *Nature* has been asking authors in the life sciences to fill out a reporting checklist since May 2013. Since June 2017, this checklist has been replaced by a reporting summary to be published as supplementary information alongside the article. While the previous checklist and current reporting summary are not uniquely intended for antibodies, these documents do ask for exhaustive identification and validation information on antibodies. Furthermore, filling out this reporting summary is a requirement for publication in *Nature.* The form needs to be uploaded at submission to prevent manuscript processing delays (article submission process tested on 09-05-2018). A more elaborate explanation of these journals' guidelines can be found in Article S1.

### *Control group*

The InCites Journal Citations Reports function of Web of Science (WoS) was used to find journals comparable to *JCN* and *Nature.* Comparable journals were chosen by selecting the first journal with a similar impact factor from the Cited Journal Data. Journals in this network are considered similar because they fit in the same research area and are part of the same citation network.

In the case of *JCN,* the most similar journal is *Neuroscience* with a similar impact factor and research category of neurosciences. The journal *Neuroscience* does not have antibody validation guidelines. No specific information about antibodies can be found in the author guidelines for article submission on *Neuroscience's* website. The website does provide information on resource materials in general. The source of all materials used should be provided and the location of each supplier should be provided on first use in the text (information from Guide for Authors section on website *Neuroscience,* retrieved 04-05-2018). However, at submission, authors are not asked to provide information on the materials they used or on antibodies specifically (article submission process tested on 09-05-2018).

A comparable journal to *Nature* is *Science*, as they are both multidisciplinary journals with similar impact factors. Using the same method as before, *Science* is also the first journal with a comparable impact factor in the list of cited journals of *Nature.* While *Science* does state that antibodies should be validated, this instruction is cursory and hard to find. It is
written between brackets on the 'Editorial Policies' page of the *Science* website, rather than in author instructions. There is no further elaboration on what validation should consist of. Furthermore, the manuscript submission process of *Science* never asks authors for validation information on materials that are used in a study and never mentions antibodies. Hence, *Science* can be regarded as having very marginal, to no guidelines on antibody reporting when compared to *Nature* (information from Information for Authors/Editorial Policies section on website *Science,* retrieved 04-05-2018; article submission process tested on 09-05-2018).

## Article selection

WoS was used to search for articles published between 2002 and 2003 or between 2017 and 2018 in each of the four journals. These lists of articles were then randomly sorted by assigning them a random number and sorting these numbers from smallest to largest. The first 15 articles in these lists in the time span of August 2002–August 2003 and June 2017–June 2018 (according to the WoS publication date) that contained antibodies as a resource material were then selected for assessment, selecting a total of 120 articles. Using antibodies as a resource material means that they are used as a tool in a biological method such as western blotting or immunohistochemistry. Antibodies can also themselves be the subject of scientific research. In this case the article was only included in the sample if antibodies were additionally used as a resource material. To determine if an antibody was used in a study, first the title of the article was used to determine if the article was published in a field that could use antibodies (e.g., life sciences and not physics). Next, the full text of the article was visually screened and screened for the word 'anti' to see if antibodies as a resource material were mentioned in the article. If this word did not occur in the full text in relation to antibodies, it was assumed that no antibodies had been used. In case of doubt, supplementary information was used to determine if an article should be included in the sample.

## Coding
### Coding protocol

All articles and their supplementary information were then scanned for antibody (validation) information, and this information was collected in a coding table. The coding table was carefully constructed by evaluating the literature for different antibody validation methods, by consulting researchers working with antibodies, and by checking the table on a set of testing articles.

Information was collected per article, not per antibody. If multiple antibodies were validated by multiple methods of validation, all methods were noted for this paper. In case of doubt about the answer to a question, the decision was always made to rule in favour of the article. For example, the highest number of validated antibodies would be written down in case of doubt about how many antibodies were validated.

The coding table consisted mostly of 'yes' and 'no' questions such as: 'are all primary antibodies validated?' or 'is validation carried out by positive control?' When a particular question was not relevant for the evaluated article (for example, the questions about type of validation when no antibodies were validated), this question was evaluated as 'not



**Figure 2** **Example of explicit antibody validation.** To test the specificity of the antibody, a comparison was made between the western blots provided by the supplier of the antibody and western blots carried out by the authors of the paper. According to the supplier, two bands should be observed: one between 25 and 37 kDa and one between 50 and 75 kDa. According to the authors, the additional band at >250 kDa is likely caused by six connexin protein subunits that form a connexin hemi-channel. The authors explicitly mention antibody validation as the purpose of this experiment: *"We tested the antibody's specificity using a western blot of Gulf toadfish whole brain homogenates."* Figure from *Rosner et al. (2018)*.

applicable'. When it was impossible to answer a question for a certain article, this question was coded 'unclear'. The answer 'unclear' was classified as missing in data analysis.

We wanted to know the percentage of articles that report validation or information on all primary antibodies from the sample of all articles that use antibodies. Therefore, for the questions 'are all primary antibodies validated?' and 'is the basic information of all primary antibodies complete?', the answer 'not applicable' was coded as 'no' to calculate the percentage of articles that contain this information in all articles with antibodies. For all other questions about validation type, the answer 'not applicable' was classified as missing. We made this choice because we wanted to know the prevalence of these validation types in articles with validated antibodies, instead of their prevalence in all articles using antibodies.

A full overview of the coding protocol, the questions in the coding table and the different antibody validation methods is provided in Article S2.

### *Implicit and explicit validation*

Because we assessed antibody validation information by reading the publication, our coding made no distinction between explicit or purposeful validation and implicit or 'accidental' validation, as the intention to validate cannot always be inferred from the text. Whether explicitly presented as validation or not, in both cases the antibody (and thus the article it was used in) was coded as validated. However, for the interpretation of our findings, it is important to understand this distinction between implicit and explicit validation.

Explicit validation means that the article provides information with the explicit purpose to attest antibody validity. In this case, experiments are presented with the purpose of validating the antibody, and/or the words antibody validation are explicitly mentioned. An example from one of the evaluated articles would be the use of western blot with molecular weight markers to validate the antibody (see Fig. 2).

**Figure 3** **Example of implicit antibody validation.** Atg7 defective cells were generated by inhibiting gene expression of the Atg7 gene with RNA interference. The cells in which Atg7 was silenced show no staining with anti-ATG7, while wild type cells do. This type of validation was marked as validation by negative control (RNA interference) and by genetic strategies of the five pillars. From *Abu-Remaileh et al. (2017)*. Reprinted with permission from AAAS.

Implicit validation here means that from reading the article, claims about the validity of the antibody can be made, but no experiments have specifically been reported to establish the validity of the antibody. The words 'antibody validation' are not mentioned. An example from one of the evaluated articles would be the use of RNA interference as an antibody validation method (see Fig. 3).

## Methodological reliability

All 120 articles were evaluated by one of us (Joyce Hoek). An independent rater (Wytske Hepkema) then assessed a randomly selected sample of these articles to determine the interrater reliability. This sample consisted of 2 articles for each journal and time frame, resulting in 16 articles or 13% of the total sample. Since Wytske Hepkema only analysed a small percentage of the total sample, we only used her answers to estimate the interrater reliability, and we used the answers of Joyce Hoek for data analysis.

The percentage of agreement between both raters (Joyce Hoek and Wytske Hepkema) was calculated using the agree function of the irr package (*Gamer et al., 2012*) in R. For nominal variables, Cohen's kappa was calculated using the cohen.kappa function of the psych package (*Revelle & Revelle, 2015*) in R. For continuous variables, the intra-class correlation coefficient (ICC) was calculated using the icc function of the irr package in R. ICC estimates were calculated based on a single-measures, consistency, two-way mixed-effects model. Missing values were not taken into account in the calculation of kappa or the ICC. Cohen's kappa was indeterminate in some cases because one or both raters answered 'yes' or 'no' for all evaluated articles. This is indicated by IND in the table. Table 1 shows the percentage agreement and Cohen's kappa or intra-class correlation coefficient.

The agreement between raters varied considerably between the different questions, from a kappa of 1 to a worse than chance kappa of $-0.20$. Antibody validation information is often not explicitly reported, as opposed to antibody identification information, which also had a higher interrater agreement. The low agreement on antibody validation questions thus illustrates the current problem of antibody validation reporting: it is very difficult for a reader to assess the validity of antibodies used in paper and therefore assess the validity of the performed experiments. A more standardized way of reporting could improve this. The large disagreement indicates that it is difficult to establish the validity of antibodies from reading the publication when antibody validation is not explicitly mentioned. The disagreement between the two raters when it comes to the categories 'validation carried out

**Table 1 Interrater reliability of article coding process.**

| Question | Percentage agreement | Cohen's kappa | 95% CI | N |
|---|---|---|---|---|
| Basic information all primary antibodies complete? | 100% | 1.00 | 1.00–1.00 | 16 |
| Any validation information present (at least one antibody)? | 75% | 0.50 | 0.13–0.87 | 16 |
| All primary antibodies validated? | 94% | IND | / | 16 |
| *If validated:* | | | | |
| Reference to validation by antibody supplier? | 100% | IND | / | 8 |
| Reference to validation in the literature? | 88% | 0.75 | 0.31–1.00 | 8 |
| Reference to validation information in database? | 100% | IND | / | 8 |
| Antibody validation carried out by authors of article? | 63% | −0.20 | −0.49–0.088 | 8 |
| *If carried out by authors, which method?* | | | | |
| Molecular weight similar to target (in WB)? | 83% | 0.57 | −0.12–1.00 | 6 |
| Spatial localization? | 100% | 1.00 | 1.00–1.00 | 6 |
| Pre-adsorption/blocking peptide? | 100% | 1.00 | 1.00–1.00 | 6 |
| Using secondary antibody without primary? | 100% | IND | / | 6 |
| One of the five pillars? | 100% | 1.00 | 1.00–1.00 | 6 |
| Positive control? | 67% | 0.33 | −0.23–0.90 | 6 |
| Negative control? | 67% | IND | / | 6 |
| Other validation method? | 50% | 0.00 | −0.75–0.75 | 6 |
| **Question** | | **ICC** | **95% CI** | **N** |
| How many antibodies were used? | | 0.995 | 0.986–0.998 | 15 |
| How many antibodies were validated? | | 0.684 | 0.263–0.886 | 14 |
| Percentage of validated antibodies? | | 0.663 | 0.226–0.878 | 14 |

by the authors of the paper', 'negative and positive control' and 'other types of validation' illustrates this. These types of validation are often consequential to the methods used in an experiment. For example, antibodies are used to validate the silencing of a gene, and, at the same time, the silencing of this gene can be seen as a method to validate the antibody. This is a type of circular reasoning that only holds if either the antibody or the silencing has proven to be valid by other methods. Therefore, it is difficult to judge if the experiment validates the antibody or not. Because of the low agreement on these questions, they were not taken into further consideration during data analysis.

On the other hand, the more explicit types of validation are easier to recognize and to judge by readers of the paper. This is illustrated by the high percentage of agreement between both raters and kappa value on the categories 'validation by reference to supplier, the literature, or a database', 'molecular weight similar to target', 'spatial localization', 'pre-adsorption' and 'secondary antibody without primary'.

## Data analysis

We compared the reporting of antibody validation information between the samples of journals with and without guidelines and between the samples of August 2002–August

2003 and June 2017–June 2018. We expected the proportion of articles reporting validation to be higher in 2017 than in 2003 in journals that implemented guidelines. Furthermore, the proportion of validation was expected to be larger in 2017 in the samples of journals that implemented guidelines compared to journals that did not implement guidelines. We measured the difference in validation reporting by comparing the percentage of validated antibodies per article and the proportion of articles reporting any validation information. We also decided to look at the proportion of articles reporting complete identification or validation information on all primary antibodies as a more robust measure of validation rigour.

For the comparisons of proportions, one-tailed $p$-values were calculated, since these hypotheses are directional. When comparing the samples of journals without guidelines between 2003 and 2017, no increase in proportion was expected. Likewise, the samples of similar journals with and without guidelines taken at baseline in 2003 were not expected to be different. For this reason, two-tailed $p$-values were calculated in these cases. Because of the small sample size, we used Fisher's exact test (fisher.test in R) to compare the two proportions (instead of the chi-square or z-test for two proportions). This function uses the conditional Maximum Likelihood Estimate as an estimate of the odds ratio. For one-tailed tests, the odds ratio goes to infinity. This is indicated by INF in the text. To account for multiple comparisons, we used the Holm-Bonferroni correction (p.adjust in R) to adjust all calculated $p$-values. Data analysis was carried out using R version 3.3.3.

In line with recent debate about the value of $p$-values (*Wasserstein, Schirm & Lazar, 2019*) and because of the exploratory nature of our study, we decided to not only look at statistically significant ($p < 0.05$) results, but also take into account meaningful but non-significant differences. To us, a meaningful difference is a substantial increase in proportion of validated articles, accompanied by a large odds ratio. We think these differences are interesting and might be good targets to further investigate if this study were to be replicated.

We provide our data and R code in Data S1 and S2 and Article S3. Our data can also be found at https://doi.org/10.17026/dans-xhk-74m4.

# RESULTS

## Percentage of validated antibodies per article

For each evaluated article, the percentage of validated antibodies per article was calculated from the total number of antibodies used and the number of validated antibodies. An overview of these numbers can be found in Table S1.

Figure 4 provides a descriptive overview of the percentages of validated antibodies in each evaluated article as well as the mean percentage per journal. This figure shows that the mean percentage of antibodies that have been validated per article for each journal slightly increased in both journals with guidelines (*JCN* and *Nature*) between 2003 and 2017. Combined, these journals increased from an average of 39% to 57% of validated antibodies per article. This percentage decreased in the journals that did not introduce reporting guidelines. These journals went from an average of 23% to 14% of validated antibodies per article.

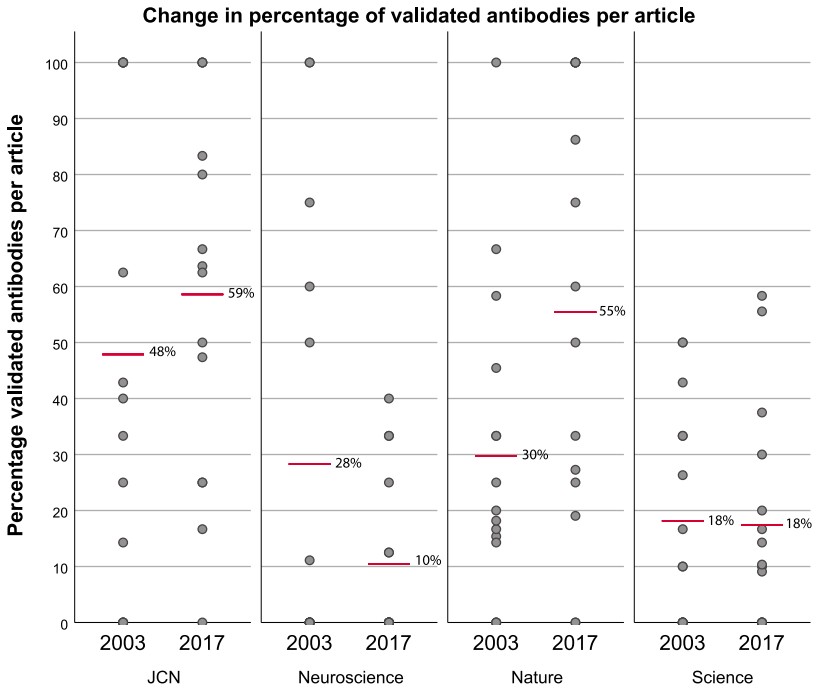

**Figure 4** **Change in percentage of validated antibodies per article before (2003) and after (2017) two of the journals (*JCN* and *Nature*) implemented antibody reporting guidelines.** Grey dots represent the percentage of validated antibodies per article, red bars indicate the mean percentage of validated antibodies per article for each journal. Sample size: *JCN* 2003 $n = 15$, 2017 $n = 14$; *Neuroscience* 2003 $n = 14$, 2017 $n = 15$; *Nature* 2003 $n = 15$, 2017 $n = 14$; *Science* 2003 $n = 15$, 2017 $n = 15$.

Furthermore, there is a difference in reporting between similar journals with and without guidelines. This difference in percentage was already present before *JCN* and *Nature* introduced guidelines, but it increased after the introduction of these guidelines. The difference in average percentage of validated antibodies between *JCN* and *Neuroscience* was 20 percentage points before the introduction of guidelines and increased to 49 percentage points after the introduction of guidelines by *JCN*. Between *Nature* and *Science*, the difference in average percentage was 12 percentage points, which increased to a difference of 38 percentage points after the introduction of guidelines by *Nature*.

## Validation of at least one antibody

Next, we made comparisons about antibody validation at the article level. Instead of looking at the percentage of validated antibodies per article, we now compared the percentage of articles that reported the validation of at least one antibody (see Tables S2 and S3). The reporting of validation improved slightly, but insignificantly, in the journals that had introduced guidelines. In addition, after the introduction of guidelines, 90% of the articles that use antibodies in journals with guidelines reported validation of at least one antibody compared to 53% of articles in journals without guidelines (OR = 7.6, 95% CI = 2.11-INF; adjusted $p = .09$, one-tailed). However, since it might be quite simple to implicitly validate at least one antibody per article, we also chose to compare the percentage of articles

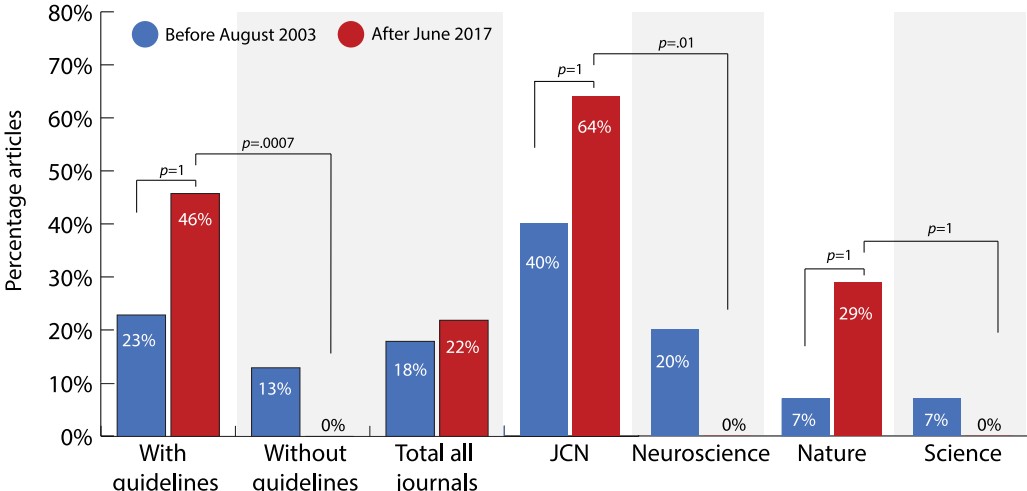

**Figure 5** **Change in percentage of articles that reported validation information on all primary antibodies used in the study before (2003) and after (2017) two of the journals (*JCN* and *Nature*) implemented antibody reporting guidelines.** Sample size: with guidelines 2003 $n = 30$, 2017 $n = 28$; without guidelines 2003 $n = 30$, 2017 $n = 30$, total 2003 $n = 60$, 2017 $n = 58$, *JCN* 2003 $n = 15$, 2017 $n = 14$; *Neuroscience* 2003 $n = 15$, 2017 $n = 15$; *Nature* 2003 $n = 15$, 2017 $n = 14$; *Science* 2003 $n = 15$, 2017 $n = 15$. P-values (one-tailed) were determined with Fisher's exact test and adjusted using the Holm-Bonferroni correction.

reporting antibody (validation) information for all primary antibodies that they used as a more robust measure of antibody reporting quality.

## Validation of all primary antibodies

First, we compared the percentage of articles that report validation information on all the primary antibodies used per study. This percentage increased between 2003 and 2017 in both journals (*JCN* + *Nature*) that implemented reporting guidelines (see Fig. 5 and Tables S4 and S5). On average, this percentage increased by 23 percentage points in the journals that had implemented guidelines (OR = 2.80, 95% CI 0.96-INF; adjusted $p = 1$, one-tailed). In contrast, the percentage of articles with validation information on all primary antibodies decreased in both journals without guidelines (*Science* + *Neuroscience*). Furthermore, there was no meaningful increase in the total sample (18% to 22%, OR = 1.28, 95% CI = 0.48–3.52; adjusted $p = 1$, two-tailed).

At the journal level, there was an increase of 24 percentage points (OR = 2.61, 95% CI = 0.60-INF, adjusted $p = 1$, one-tailed) in articles that reported validation information on all primary antibodies in *JCN* and an increase of 22 percentage points in *Nature* (OR = 5.3, 95% CI = 0.59-INF; adjusted $p = 1$, one-tailed).

By comparing the journals with guidelines to the journals without guidelines, we can see that the difference in reporting between these similar journals in our sample substantially increased after the introduction of guidelines. After June 2017, 46% of the articles in journals that did implement guidelines (*JCN* + *Nature*) reported validation of all primary
antibodies after the introduction of guidelines, compared to 0% in journals that did not implement guidelines (*Neuroscience + Science*), a difference of 46 percentage points (OR = INF, 95% CI = 6.5-INF; adjusted $p$ = .0007, one-tailed). While a difference in percentage was already present before the introduction of guidelines in 2003, this difference was much smaller (10 percentage points, OR = 1.96, 95% CI = 0.43–10.3; adjusted $p$ = 1, two-tailed).

At the journal level, there was a large difference of 64 percentage points (OR = INF, 95% CI = 5.1-INF, adjusted $p$ = .01, one-tailed) in validation reporting between the journals *JCN* and *Neuroscience* after June 2017. Between *Nature* and *Science* there was a difference in reporting of 29 percentage points (OR = INF, 95% CI = 1.09-INF; adjusted $p$ = 1, one-tailed) after *Nature* had introduced guidelines, a difference that was not present before.

## Identification of all primary antibodies

In addition to looking at antibody validation, we also evaluated how well antibody identification information was reported for all primary antibodies used in a study. Antibodies are considered identifiable if enough information is provided in the publication for readers to be able to track down the antibody and obtain it themselves. In order to do so, either an RRID or the name, supplier and catalogue number of a commercial antibody needs to be reported. For non-commercial antibodies either an RRID or the host-animal and immunogen used needs to be reported. Figure 6 and Tables S6 and S7 show the percentage of articles that reports identification information on all primary antibodies that were used in the article.

The percentage of articles that reported identification information on all primary antibodies increased considerably between 2003 and 2017 in journals that implemented antibody guidelines. Overall, the percentage of articles that reported identification information on all primary antibodies increased with 58 percentage points from 10% to 68% (OR = 17.8 95% CI = 4.8-INF; adjusted $p$ = .0003, one-tailed) in journals with guidelines (*JCN + Nature*). Of the separate journals, *JCN* showed the most improvement in reporting. After the implementation of guidelines, its percentage increased from 20% to 93% of articles that use antibodies (OR = 45.2, 95% CI = 5.5-INF; adjusted $p$ = .003, one-tailed). In the journals *Neuroscience* and *Science* this percentage also increased, but the increase was much smaller than in the journals with guidelines. Overall, the percentage of articles reporting identification information on all primary antibodies increased with 18 percentage points from 10% to 28% (OR = 3.36, 95% CI = 0.70–22.1; adjusted $p$ = 1, two-tailed) in journals without guidelines (*Neuroscience + Science*). Apart from that, the overall reporting of identification information improved over time. For all journals combined this percentage increased by 37 percentage points (OR = 7.9, 95% CI = 2.81–26.3; adjusted $p$ = .0005, two-tailed). This suggests that other factors such as an increase in awareness, the availability of identifying information, or a changing view on the importance of antibody reporting by the scientific community might also play a role in how well antibody identification information is reported.

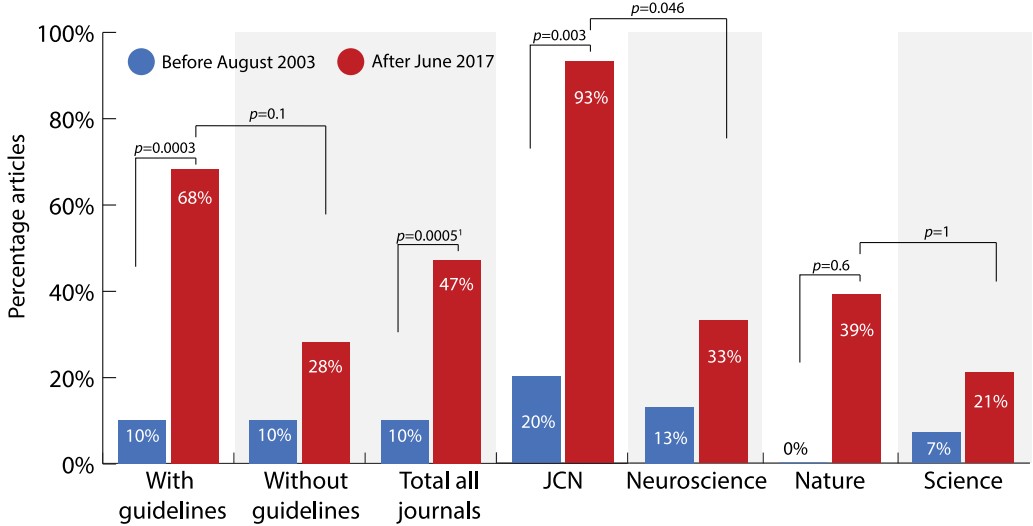

## Identification information - all primary antibodies

**Figure 6** **Change in percentage of articles that reported identification information on all primary antibodies used in the study before (2003) and after (2017) two of the journals (*JCN* and *Nature*) implemented antibody reporting guidelines.** Sample size: with guidelines 2003 $n = 30$, 2017 $n = 28$; without guidelines 2003 $n = 30$, 2017 $n = 29$; total all journals 2003 $n = 60$, 2017 $n = 57$; *JCN* 2003 $n = 15$, 2017 $n = 15$; *Neuroscience* 2003 $n = 15$, 2017 $n = 15$; *Nature* 2003 $n = 15$, 2017 $n = 13$; *Science* 2003 $n = 15$, 2017 $n = 14$. *P*-values were determined with Fisher's exact test and adjusted using the Holm-Bonferroni correction. [1]two-tailed *p*-values, other *p*-values are one-tailed.

## Change in type of validation

If an article reported validation of at least one antibody, the types of validation that were carried out in the article were further specified. An article can contain multiple types of antibody validation, since one antibody can be validated in multiple ways, or multiple antibodies can be validated by different methods. First, a distinction was made between validation by the authors of the paper themselves and validation by means of a reference to the literature, to the information of the supplier, or to validation information in a database. Validation by the authors implies an experimental check in the laboratory and involves material verification, while the other forms rely on documentation. Reference to the supplier implies a reliance on supplier information about validity, reference to the literature implies a documentation of similar use of these antibodies by other researchers, which is also possible via reference to a database.

Example of validation by the authors of an evaluated paper:

- **Independent antibody strategies of five pillars.** *"Our immunostaining in the mouse embryo cerebellum produced a similar labeling pattern as the other FoxP2 antibody"* (*Vibulyaseck et al., 2017*).

Examples of validation by reference to a third party from the evaluated articles:

- **Reference to supplier.** *"The rabbit polyclonal anti - FoxP2 antibody (AP5753b, Abgent, San Diego, California, CA, USA) produced a single major band of 80 kDa in the mouse heart tissue lysates in the manufacturer's western blot analysis."* (*Vibulyaseck et al., 2017*).
- **Reference to the literature.** *"The goat polyclonal anti - EphA4 antibody (AF641, R&D Systems, Minneapolis, MN, USA) recognized a single band of 110 kDa in HEK293 cell lysate transfected with the EphA4 gene by western blot* (*Hashimoto et al., 2012*). *Its immunohistochemical reactivity has been eliminated by preabsorption with the recombinant EphA4 protein* (*Rosas et al., 2011*)*"* (*Vibulyaseck et al., 2017*).

Figure 7 and Tables S8 and S9 show the changes in the prevalence of these types of validation before and after the introduction of guidelines in articles that contain at least one validated antibody. The percentage of articles that use validation by the authors seems extremely high. It went from 87% to 85% in journals without guidelines and from 94% to 94% in journals with guidelines between 2003 and 2017. However, these percentages might be misleading. These high percentages were often caused by a small number of validated antibodies per article, which is explained by our methodology of evaluating validation per article instead of per antibody. Moreover, this method of validation is often implicitly carried out as a consequence of the experimental method being used. Since the reliability of the rating of this category is questionable ($\kappa = -0.20$), we did not perform further statistical analysis on this category.

Both the use of the validation methods 'reference to the literature' and 'reference to the antibody supplier' increased in journals with guidelines (*JCN + Nature*) after the introduction of these guidelines. These methods increased by 24 percentage points (OR = 2.62, 95% CI = 0.86-INF; adjusted $p = 1$, one-tailed) and 27 percentage points (OR = INF, 95% CI = 1.96-INF; adjusted $p = 0.4$, one-tailed) respectively.

Before guidelines were installed, there was a small difference in the prevalence of these validation types between journals with and without guidelines. This difference increased after the introduction of guidelines. Before the introduction of guidelines, there was a difference of 13 percentage points (OR = 1.99, 95% CI = 0.36–14.4; adjusted $p = 1$, two-tailed) in validation by reference to the literature between journals with and without guidelines. After the introduction of guidelines, this difference increased to 43 percentage points (OR = 8.3, 95% CI = 1.8-INF; adjusted $p = 0.3$, one-tailed). Similarly, there was no difference in the use of validation by reference to the antibody supplier before the introduction of guidelines. However, after guidelines had been installed, there was a difference of 27 percentage points (OR = INF, 95% CI = 1.34-INF; adjusted $p = 1$, one-tailed) between journals with and without guidelines. It is notable that validation by reference to information provided in an online antibody database is not used at all.

## Change in type of validation carried out by authors of paper

If an antibody was experimentally validated by the article's authors, we further specified what type of validation was carried out. Figure 8 and Tables S8 and S9 show an overview of the change in reported validation types. Figure 8 contains methods that are mostly used to validate antibodies explicitly. Other types of (often more implicit) validation,

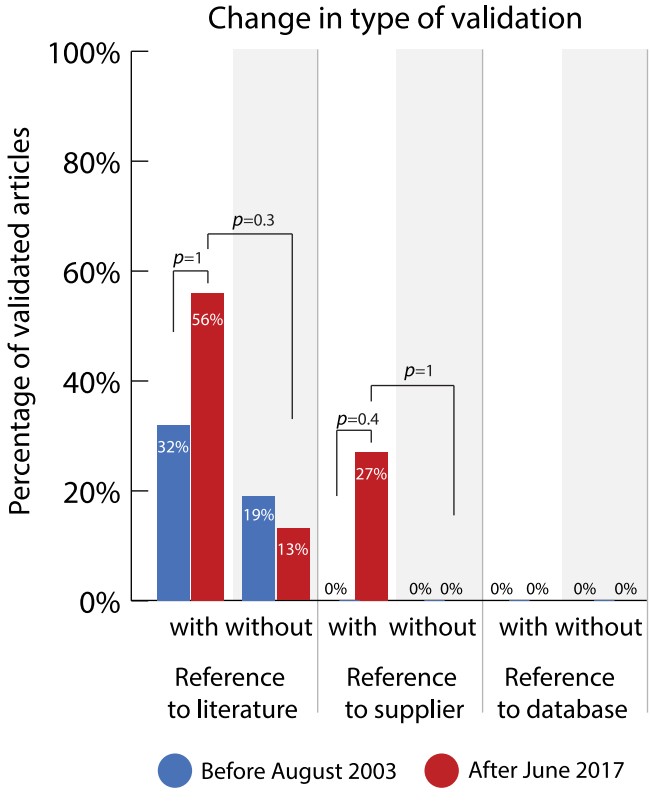

**Figure 7** **Change in percentage of validated articles reporting different types of validation before (2003) and after (2017) the journals with guidelines implemented antibody reporting guidelines.** Sample size (identical for all types of validation unless stated otherwise): with guidelines 2003 $n = 23$, 2017 $n = 27$; without guidelines 2003 $n = 16$, 2017 $n = 16$; exceptions: reference validation literature with guidelines 2003 $n = 22$; reference validation supplier with guidelines 2017 $n = 26$. P-values (one-tailed) were determined with Fisher's exact test and adjusted using the Holm-Bonferroni correction.

such as positive and negative control, were also registered but not taken into account for comparisons because of low interrater reliability.

Examples of validation from the evaluated articles:

- **Pre-adsorption.** *"In control experiments, primary antibody was pre-adsorbed with rat ANP (Bachem-Peninsula Laboratories; 1 μM overnight at 4 °C) prior to incubation with tissue sections to assess the degree of non-specific staining. Under these conditions, low levels of background staining were observed in cortex"* (*Wiggins, Shen & Gundlach, 2003*).
- **Spatial localization.** *"(…) a polyclonal antisera against rat ANP were used in an attempt to visualize ANP-like-IR in the cerebral cortex and to determine the effect of CSD on its level and cellular distribution. Consistent with previous reports, ANP-like-IR was consistently detected in subcortical regions, with a high density of nerve-fibre staining found throughout areas such as the bed nucleus of the stria terminalis (…), the preoptic hypothalamus, areas of the amygdala and in the paraventricular thalamic nucleus (see Kawata et al., 1985, Skofitsch et al., 1985)"* (*Wiggins, Shen & Gundlach, 2003*).

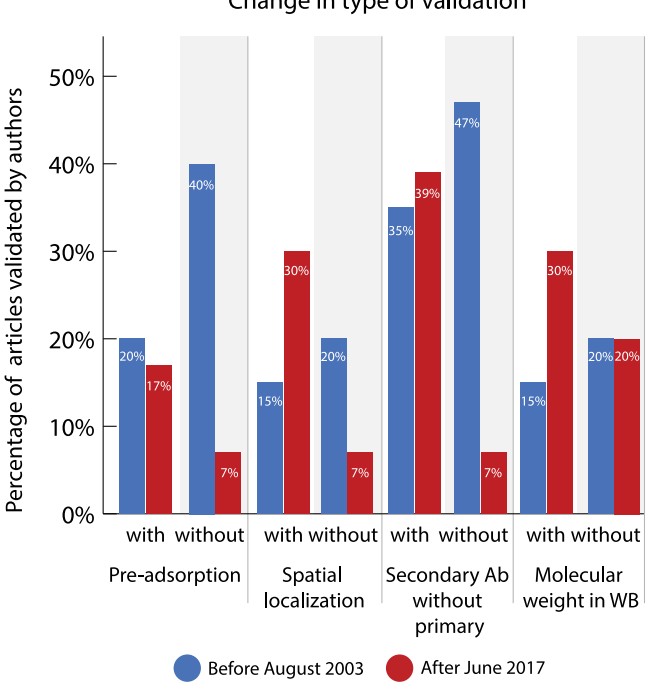

Figure 8 **Change in percentage of validated articles reporting different types of validation before (2003) and after (2017) the journals with guidelines implemented antibody reporting guidelines.** Sample size (identical for all types of validation): with guidelines 2003 $n = 20$, 2017 $n = 23$; without guidelines 2003 $n = 15$, 2017 $n = 15$. P-values were determined with Fisher's exact test and adjusted using the Holm-Bonferroni correction. All adjusted p-values are 1.

- **Secondary antibody without primary.** *"Additional control incubations with primary antibody omitted were routinely included and resulted in a lack of specific staining"* (*Wiggins, Shen & Gundlach, 2003*).
- **Molecular weight in western blot (and KO).** *"The rat monoclonal anti - Pcdh10 antibody (MABT20 clone 5G10, Millipore, Billerica, MA, USA) produced a single band of 137 kDa in the P0 mouse cerebellum and whole brain in our western blot. The band disappeared completely in samples from Pcdh10 - KO mice (OL - KO homozygotes)"* (*Vibulyaseck et al., 2017*).

Before the introduction of guidelines, all types of validation were used more often in journals without guidelines. However, after the introduction of guidelines, the use of these validation methods decreased in these journals without guidelines while there was a slight, but nonsignificant, increase of their use in the journals with guidelines. This means that the differences in the types of validation that were mostly used increased between journals with guidelines and similar journals without guidelines. After the introduction of guidelines, there was a difference of 10 percentage points in the use of pre-adsorption (OR = 2.87, 95% CI = 0.33-INF; adjusted $p = 1$, one-tailed), 23 percentage points in the use of spatial localization (OR = 5.9, 95% CI = 0.80-INF; adjusted $p$-value=1, one-tailed), 32 percentage points in the use of a secondary antibody without a primary one (OR = 8.6, 95% CI =

1.22-INF; adjusted $p = 1$, one-tailed), and 10 percentage points in the use of molecular weight in western blot (OR $= 1.73$, 95% CI $= 0.39$-INF; adjusted $p = 1$, one-tailed) between similar journals.

Interestingly, there is a difference in how often these validation types are used in the two journals that introduced guidelines. In the journal *Nature*, the methods pre-adsorption and spatial localization were not used at all before and after the introduction of guidelines. The small increase in reporting of spatial localization is thus solely caused by its use in the journal *JCN*.

## DISCUSSION

These results suggest that journal guidelines might have some positive influences on antibody validation reporting. For example, this is suggested by the large difference in the percentage of articles reporting validation information of all primary antibodies between journals with and without guidelines after the introduction of these guidelines. However, it is notable that in most of our analyses a difference in reporting was already present between the four journals before the introduction of guidelines. This could suggest that the journal itself (e.g., its editorial attention to validation, or earlier publications about validation), regardless of the presence of reporting guidelines, already has an influence on antibody validation reporting. Another explanation might be that there is a difference in validation practices and journal selection between research areas or groups.

Moreover, our results provide some insight in how journal guidelines might work. This can be illustrated by comparing the increase in reporting of validation information with identification information of all primary antibodies. On average, the reporting of identification information improved by 58 percentage points, while the reporting of validation information only improved by 23 percentage points in journals that introduced guidelines. This suggest that the introduction of antibody reporting guidelines particularly affected aspects of antibody reporting that are relatively easy to change. Adding identification information for each antibody used in your publication, such as catalogue numbers, is easier than providing evidence of actual validation. From an editorial perspective, the provision of identity information such as catalogue numbers is also easier to check. At least for some of the articles, researchers seem to have opted for the easier solutions to comply with antibody guidelines by providing readily available information.

A similar indication for solutions that choose the 'path of least resistance' to comply with guidelines, can be found in the use of third-party validation information. Once again, this information is relatively easy to include in a paper with a simple reference and requires no extra experimental work. Our results show that validation by reference to the literature increased by 24 percentage points in journals that introduced guidelines, while this percentage decreased in journals without guidelines. Likewise, validation by reference to the antibody supplier's information increased by 27 percentage points in journals with guidelines, while this method was never used in journals without guidelines. These results are not significant, but the odds ratios seem to suggest (especially in the case of reference

to supplier's information) that these validation practices merit further investigation. More research assessing how researchers regard journal guidelines, how these guidelines affect their practices, and a replication of this study with a larger sample size is recommended.

Because the guidelines mostly seem to influence aspects of antibody reporting that are relatively easy to provide, we concluded that the effects of journal guidelines on antibody reporting are limited. This limited effect of journal guidelines is consistent with findings in similar research on journals' measures to improve reporting about research resources. A recent evaluation of the 2013 *Nature* life science checklist shows that it improved the reporting of some items, but also that approximately half of the compliant articles did not actually use the tools recommended by the checklist (*Han et al., 2017*). Another evaluation of the *Nature* checklist showed improvements in transparency of reported information, but much more modest improvements in experimental design. Furthermore, the study found that the checklist did not improve antibody reporting (*NPQIP Collaborative group, 2019*). However, with regard to antibody identifiability, our results are more optimistic than previous studies. Vasilevsky et al. found no relationship between reporting guidelines and identifiability of resource materials in biomedical research. Identifiability of resources was actually higher in journals with no or loose guidelines than in journals with strict guidelines. Antibody identifiability in the *Journal of Comparative Neurology* was only slightly higher than average across all journals (*Vasilevsky et al., 2013*). While our results point towards some improvements, it is clear that journal guidelines are not a definitive solution to fix validation and misidentification problems.

Much seems to depend on how journal guidelines are implemented. Mere symbolic support for guidelines in editorial instructions may not be enough to improve reporting practices. Baker et al. found that the endorsement of the ARRIVE (Animal Research: Reporting of In Vivo Experiments) guidelines by journals had little to no impact the reporting of animal studies in these journals (*Baker et al., 2014*). *Hair et al. (2019)* found that even requiring authors to complete an ARRIVE checklist after submission of their manuscript did not improve compliance. In another study of journal guidelines regarding RRIDs, compliance was the lowest when the journals only provided author instructions. Compliance became higher when authors received mailed instructions, and it became very high when editorial staff assisted authors with their RRIDs (*Bandrowski et al., 2016*). These results suggest that journal guidelines may have a beneficial effect, but that they require additional measures to make them effective.

Another interesting observation from our study is that, in contrast to the reporting of antibody validation, the reporting of information on the identity of all primary antibodies increased in all four journals between 2003 and 2017. Moreover, this percentage increased significantly by 37 percentage points in the total sample of all four journals combined. Although the improvement was larger in journals that introduced guidelines, it is likely that this improvement is at least partly caused by a growing awareness of antibody problems. That this growing awareness only seems to have an effect on improving antibody identification reporting and not on antibody validation reporting might be due to the current attention for reproducibility in science. Although antibody validity is important for the overall validity of research, detailed reporting of which antibodies were used is

important to ensure reproducibility. However, as we have argued, reproducible research is not necessarily valid research. We would therefore urge not to forget the validity question in the replication crisis debate. Furthermore, it would be interesting to study researchers' motivation for this change in reporting behaviour to see whether and how it relates to the reproducibility crisis.

## Limitations

Our study has several noteworthy limitations. As an exploratory study, our study was not pre-registered, which we would recommend for possible future confirmatory replications of our work. Furthermore, the labour-intensive nature of scrutinising publications for validation information means our sample and interrater testing are limited. Together with a conservative statistics test (Fisher's exact), this might have increased our Type II error. Having a low power can have several consequences for the reliability of results, such as an overestimation of the effect size. Readers of our study should be cautious of this limitation while interpreting our results.

In addition, coding of articles inevitably implies some level of arbitrary convention and interpretation, for example as we decided to code on the level of articles and not individual antibodies. No distinction was made between implicit and explicit validation, giving the researchers the benefit of the doubt with respect to their validation efforts, which may overestimate the beneficial effect of guidelines and how often certain validation methods are used. The room for interpretation about whether specific forms of experimental validation were actually used, is also illustrated by the lower agreement between raters in this respect. This again means that our results should be regarded with caution and a replication of this study with a larger sample size and multiple assessors is recommended. Because of these limitations, we tried to be as open as possible about the decisions we made during the coding process, by illustrating these choices with examples from the evaluated articles in this manuscript, and by providing an elaborate explanation of our choices in the coding table.

Last, our pseudo-experimental research design has some limitations. Because of the substantial time period from which we took our samples (2003 and 2017), the time between the intervention of implementing the guidelines and the measurement of changed behaviour is stretched. We chose this approach because of the implementation of the guidelines by the two journals at different times. However, another way to approach this could be by sampling over time, for example by using interrupted time series analysis. Moreover, the quasi-experimental design makes the effect of journal guideline introduction hard to isolate from wider influences, such as a growing awareness of the urgency of antibody validation, or the growing knowledge stored in databases such as the Antibodyregistry. Fortunately, our results do show signs of an overall growing awareness of antibody problems, since the reporting of antibody identification information significantly improved in the entire sample.

## CONCLUSIONS

This study suggests that the implementation of antibody reporting guidelines by a journal might have some positive influences on the reporting of antibody validation information, as we hypothesized. The percentage of validated antibodies per article, the percentage of articles which reported validation and identification of all primary antibodies, and different types of validation all increased in journals with guidelines after the introduction of these guidelines. However, improvements were particularly visible for forms of antibody information that are relatively easy to provide, such as providing complete identification information. Strictly speaking, such information supports the reliability of antibodies, and it is conceivable that securely identified and previously validated antibodies can be used reliably in identical applications. Nevertheless, improvements of more robust experimental validation were modest, at best. Combined with the results of previous studies on journal guidelines, this suggests that the effect of journal antibody guidelines by themselves may be limited and may require additional measures to ensure effective implementation.

In light of the quasi-experimental and exploratory nature of our study, we need to be careful with drawing these conclusions. Other factors, such as a general shift in research practices, a change in publication behavior, a preference for publishing in a journal that adheres to the values of a researcher, or changed editorial practices might also play a role. More research would therefore be needed to study how researchers make decisions with regard to antibody validation and how they adjust their research practices in response to changing journal guidelines.

## ACKNOWLEDGEMENTS

We would like to thank Jeffrey Stuart, Nicole Vasilevsky, Malcolm MacLeod and one anonymous reviewer for their constructive feedback on our manuscript; Ger Pruijn, Serge Horbach, Freek Oude Maatman, Merle-Marie Pittelkow, Jasmine Muradchanian and Ymkje Anna de Vries for comments; and the Research Quality Team for inspiring discussions.

### Funding

The authors received no funding for this work.

### Competing Interests

The authors declare there are no competing interests.

### Author Contributions

- Joyce M. Hoek conceived and designed the experiments, performed the experiments, analyzed the data, prepared figures and/or tables, authored or reviewed drafts of the paper, and approved the final draft.
- Wytske M. Hepkema performed the experiments, authored or reviewed drafts of the paper, and approved the final draft.

- Willem Halffman conceived and designed the experiments, authored or reviewed drafts of the paper, and approved the final draft.

## Data Availability

We provide our data and R code in the supplementary files. Our data can also be found at 10.17026/dans-xhk-74m4.

## Supplemental Information

Supplemental information for this article can be found online at http://dx.doi.org/10.7717/peerj.9300#supplemental-information.

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
