# Peer review of "The effect of journal guidelines on the reporting of antibody validation"

_PeerJ, doi:10.7717/peerj.9300_

## Round 0.1 · original submission · Major Revisions

Three reviewers have relatively disparate points of view on your manuscript. All of their comments should be addressed in your resubmission.

·

Basic reporting

This is a great study that evaluates the reporting of antibody validation in studies that are published in journals that require validation in their instructions to authors.

As a researcher who previously worked with antibodies during my time on the bench (western blots and flow cytometry), I understand the fickle nature of antibodies as reagents, and the importance in validating antibodies for use in scientific assays.

This paper nicely describes the background and previous studies related to this topic, the literature is well-referenced and cited.

The writing is clear and unambiguous.

The figures are very clear and professionally done, the raw data is shared. Please include a data dictionary to describe all column headers in the supplemental data.

Experimental design

The research question is well defined and very relevant to current issues in scientific reproducibility (ie the reproducibility crisis).

The execution of this study is well done. The methods are sufficiently reported and this study should be reproducible based on the reporting and supplemental data.

I recommend sharing your data in a public repository, in addition to sharing it as supplemental data.

Validity of the findings

The findings seem valid and robust. The conclusions are well supported and the discussion does a nice job of pointing out the limitations of the study.

Additional comments

It is encouraging to see that the percentage of authors who report validation studies for their antibodies has increased over time. To be honest, I would have expected these numbers to be higher, but you give good reasons in the paper for the lack of adherence to the policy.

I hope that more journals adopt these policies and find ways to enforce them. I hope there is a shift in the way research is performed, to recognize the importance in validating antibody reagents prior to performing experiments. Antibodies are valuable reagents for research, but the lack of reliability and inconsistency in their results is really a detriment to science.

This study is a nice contribution to the scientific community and I hope it will push further change, on the part of journals and researchers, to improve the validation of antibodies used in research.

·

Basic reporting

The work is well presented, nad reported well according to the four criteria given

Experimental design

This is a critically important issue ... it is becoming increasingly apparent that using the wrong antibody, or the right antibody in the wrong way, or insufficient reporting of the antibodies used, is one of the major drivers certainly of difficulties in attempted replication and likely also of findings which do not replicate.

I have some experience of conducting a similar (but much more shallow) review of changes in reporting at Nature journals, so I understand at least some of the challenges.
1. We analysed over 800 manuscripts, with each manuscript assessed independently by two assessors, and differences resolved by a third. The reason we did this many was because our sample size calculations indicated that this large number would be required to detect a significant improvement in reporting. No power calculations are offered in the current manuscript, which is a weakness.
2. We are not told whether the approach, or the statistical analysis plan, were determined in advance. If this is not the case, it is possible that the approach taken might have changed - consciously or unconsciously - in response to emerging data. This is a weakness.
3. The disagreement between assessors - based on a small sample of the total - is substantial. Either the first assessor is close to perfect and so this doesn't matter, or the complexities of what is being assessed (and I agree it is complex indeed, and these papers can be surprisingly difficult to read) mean that one assessor is not enough. Our more recent evaluation of the MDAR framework where in-house editors tried to apply the tool (see https://osf.io/2k3va/) suggests that disagreement is the norm, and that two assessors (with reconciliation from a third) would give more reliable information.
4. The time periods over which change was sampled were substantial - in this quasi-experimental setting we rely a little on Bradford Hill criteria, of which timeliness is critical (the change in behaviour followed the intervention). With a stretched timeline, this is less convincing. An alternative might be to sample over time, with an interrupted time series analysis or even a quality improvement tool such as a control chart.

Validity of the findings

See my concerns above. I appreciate the provision of the underlying data, and the characterisation of this work as relatively early (more exploratory than hypothesis testing), but I do think this could very usefully form the basis for the design and pre-registration of an adequately powered study.

Additional comments

I think this is a very worthwhile endeavour. However, to provide sufficient motivation to overcome the inertia which resists change, I think we need more convincing evidence.

Reviewer 3 ·

Basic reporting

No comment.

Experimental design

No comment.

Validity of the findings

No comment.

Additional comments

The manuscript is of good quality and interesting. Perhaps some condensation would make the article more readable. I do not agree with the notion that the guidelines of the journals "might have some limited influences". The general interpretation seems to be too negative compared to the results reported. From my point of view, the influence of the guidelines was surprisingly large. One has to consider that the sample size was relatively small and long term effects were not examined. The significant differences between the raters also show that the extraction ("coding") of the data from the written text is by far not trivial and unambiguous. This also can be suspected for any referees and the journal editors, who have to decide, whether the manuscript meets the validation guidelines or not. I would conclude from these results that the effect of explicit guidelines is significant and that some standardization is needed to facilitate the examination of the validation level and in the ideal case might make these data even machine-readable.

---

## Round 0.2 · Minor Revisions

There is just one item to consider and respond to. Please address it and resubmit when you are ready.

·

Basic reporting

no further comments

Experimental design

no further comments

Validity of the findings

no further comments

Additional comments

no further comments

Reviewer 3 ·

Basic reporting

No comment

Experimental design

No comment

Validity of the findings

No comment

Additional comments

I think that the manuscript is fine now. I do not agree with all points with the authors, however, I respect their opinions. One major objection is left, however. In this article, the authors point out that mainly antibody identification is improved by the guidelines but not many experimental validation efforts. This might be correct. I disagree, however, that this needs to be a validation deficit. If an unequivocal identification of a previously validated antibody has been performed, this antibody can be accepted as validated in all future papers using exactly the same protocol for the same application. Some antibodies have been validated many times. In my opinion, such routine applications of antibodies or assays are quite frequent. I would not demand any further validation if there is not a specific reason for that. Therefore, you should introduce a third category, besides explicit and implicit validation, which might be termed traceable or previous validation. In this case, only an unequivocal identification (and perhaps citation of original work, where the validation can be found) would be required. If this is seen as a fully acceptable validation approach, your appraisal of antibody reporting guidelines would be more positive.

---

## Round 0.3 · accepted · Accept

Thank you for your submission and attention to reviewers' comments.